# KLay: Accelerating Arithmetic Circuits for Neurosymbolic AI

**Jaron Maene & Vincent Derkinderen**
Department of Computer Science
KU Leuven
Leuven, Belgium
`{jaron.maene,vincent.derkinderen}@kuleuven.be`

**Pedro Zuidberg Dos Martires**
Centre for Applied Autonomous Sensor Systems
Örebro University
Örebro, Sweden
`pedro.zuidberg-dos-martires@oru.se`

## Abstract

A popular approach to neurosymbolic AI involves mapping logic formulas to arithmetic circuits (computation graphs consisting of sums and products) and passing the outputs of a neural network through these circuits. This approach enforces symbolic constraints onto a neural network in a principled and end-to-end differentiable way. Unfortunately, arithmetic circuits are challenging to run on modern tensor accelerators as they exhibit a high degree of irregular sparsity. To address this limitation, we introduce knowledge layers (KLay), a new data structure to represent arithmetic circuits that can be efficiently parallelized on GPUs. Moreover, we contribute two algorithms used in the translation of traditional circuit representations to KLay and a further algorithm that exploits parallelization opportunities during circuit evaluations. We empirically show that KLay achieves speedups of multiple orders of magnitude over the state of the art, thereby paving the way towards scaling neurosymbolic AI to larger real-world applications.

## 1 Introduction

Interest in neurosymbolic AI (Hitzler & Sarker, 2022) continues to grow as the integration of symbolic reasoning and neural networks has been shown to increase reasoning capabilities (Yi et al., 2018; Trinh et al., 2024), safety (Yang et al., 2023), controllability (Jiao et al., 2024), and interpretability (Koh et al., 2020). Furthermore, neurosymbolic methods often require less data by allowing a richer and more explicit set of priors (Diligenti et al., 2017; Manhaeve et al., 2018).

However, as the computational structure of many neurosymbolic models is partially dense (in its neural component) and partially sparse (in its symbolic component), efficiently learning neurosymbolic models still presents a challenge (Wan et al., 2024). So far, the symbolic components of these neurosymbolic models have struggled to fully exploit the potential of modern GPUs or TPUs.

Our work focuses on a particular flavor of neurosymbolic AI, pioneered by Xu et al. (2018) and Manhaeve et al. (2018), which performs probabilistic inference on the outputs of a neural network. This is achieved by encoding the symbolic knowledge using arithmetic circuits. While arithmetic circuits are end-to-end differentiable, they also pose certain challenges. In particular, arithmetic circuits are ill-suited to be evaluated in terms of dense tensor operations due to their high degree of irregular sparsity. In this work, we address the challenge of optimizing neurosymbolic architectures to efficiently leverage available widely available hardware. To this end, we present KLay: a new data structure representing arithmetic circuits as **k**nowledge **lay**ers which can exploit the parallel compute present in modern GPUs or TPUs.

The main advantage of KLAY is that it reduces arithmetic circuit evaluations to index and scatter operations – operations already present in popular tensor libraries. This allows for the embarrassingly parallel nature of these knowledge layers to be harnessed. Importantly and in contrast to alternative approaches that speed up sparse neurosymbolic computation graphs (Dadu et al., 2019; Liu et al., 2024), we forego the need for custom hardware-specific implementations. This makes KLAY completely agnostic towards the underlying hardware. By leveraging the compiler stacks of open-source tensor libraries, KLAY can furthermore considerably outperform existing hand-written CUDA kernels.

## 2 ARITHMETIC CIRCUITS AND NEUROSYMBOLIC AI

The symbolic knowledge in neurosymbolic AI is commonly specified as Boolean logic. *Boolean circuits* are a compact representation of Boolean logic using directed acyclic graphs (Darwiche, 2021). More specifically, the leaves in Boolean circuits correspond to Boolean variables (or their negation), while inner nodes are either $\wedge$-gates or $\vee$-gates. We make the usual assumption that Boolean circuits do not contain negation, known as *negation normal form* (NNF). Figure 1a contains an example of a Boolean circuit. A circuit can be evaluated for a set of inputs by a simple post-order traversal of the graph, meaning children get evaluated before their parents. More formally, the value $v(n)$ of a node $n$ is a defined as

$$v(n) = \begin{cases} l_n & \text{if } n \text{ is a leaf node,} \\ \bigwedge_{c \in \mathcal{C}_n} v(c) & \text{if } n \text{ is a } \wedge\text{-gate,} \\ \bigvee_{c \in \mathcal{C}_n} v(c) & \text{if } n \text{ is a } \vee\text{-gate.} \end{cases} \tag{1}$$

We write $l_n$ for the Boolean input value of the leaf node $n$ and $\mathcal{C}_n$ for the set of children of an inner node $n$. The evaluation of a circuit is a function $\mathbb{B}^N \to \mathbb{B}$ which maps the Boolean inputs of the leaves to the value of the root node.

In order to render Boolean circuits useful in the context of neurosymbolic AI, we first need to perform a so-called knowledge compilation step (Darwiche & Marquis, 2002). This compilation step transforms an NNF circuit into *deterministic decomposable negation normal form* (d-DNNF). *Determinism* (Darwiche, 2001b) and *decomposability* (Darwiche, 2001a) are two properties that guarantee certain computations can be performed tractably, such as finding the number of satisfying assignments. We refer the reader to Vergari et al. (2021) for an in-depth discussion on tractable computations on circuits. Figure 1b contains the d-DNNF circuit resulting from knowledge-compiling the circuit in Figure 1a.

The important advantage of d-DNNF circuits over general NNF circuits is that they allow linear time probabilistic inference. Assume first that the Boolean variables are not deterministic anymore but constitute Bernoulli random variables. We can then compute the probability of the d-DNNF circuit evaluating to true under the input distribution by labeling the leaves of the circuit with the probabilities of the Boolean variables and replacing $\wedge$- and $\vee$-gates with $\times$ and $+$ operations, respectively. The resulting circuit is also called an arithmetic circuit (Darwiche, 2003) and is displayed in Figure 1c.

While probabilistic inference on d-DNNF circuits has linear time complexity, transforming a NNF circuit into d-DNNF is #P-hard (Valiant, 1979). However, once the d-DNNF structure is obtained we can re-evaluate the circuit with different probabilities for the Boolean variables in the leaves. This approach has been very successful in probabilistic inference for a variety of models such as Bayesian networks (Chavira & Darwiche, 2008) and more general probabilistic programs (Fierens et al., 2015).

This compile once and evaluate often paradigm has also gained traction in neurosymbolic systems (Manhaeve et al., 2018; Ahmed et al., 2022b; De Smet et al., 2023). The high-level idea behind these approaches is to compile the symbolic knowledge once into an arithmetic circuit and let a neural network predict the probabilities to be fed into the arithmetic circuit. Given that the arithmetic circuit consists only of sum and product operations, the resulting computation graph (neural network + arithmetic circuit) is end-to-end differentiable and the parameters can be optimized using standard gradient descent methods. Furthermore, as gradient descent is an iterative method, the circuit (and its gradient) needs to be evaluated repeatedly. This yields a convenient return on investment for the

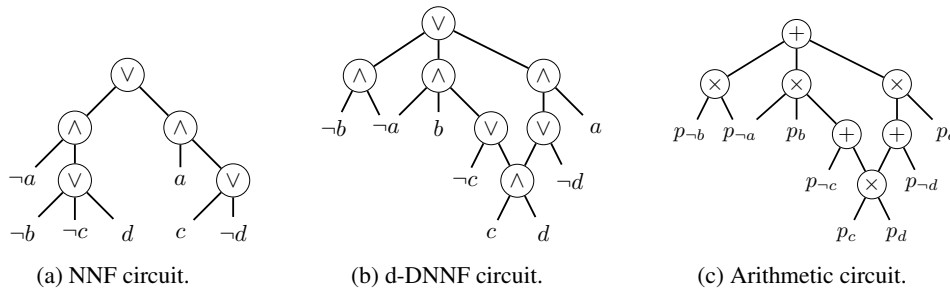

(a) NNF circuit.  (b) d-DNNF circuit.  (c) Arithmetic circuit.

Figure 1: Example of (a) a Boolean circuit, (b) its d-DNNF version, and (c) the corresponding arithmetic circuit. The arithmetic circuit replaces Boolean variables with their probabilities. Consequently, the arithmetic circuit computes the probability of the underlying Boolean circuit being satisfied. In a neurosymbolic setting, the input probabilities are predicted by a neural network.

initial expensive knowledge compilation step, which now amortizes over multiple evaluations. Appendix D explains in more detail how neural networks and circuits can be used together on a simple example.

As discussed in the introduction, arithmetic circuits currently hinder the efficient application of neurosymbolic methods as these circuits are not well-suited to be evaluated as parallel tensor operations. As a matter of fact, the standard way to evaluate an arithmetic circuit in a neurosymbolic context is to naively evaluate every node one by one (Manhaeve et al., 2018; Ahmed et al., 2022a;b) using a naive traversal of the arithmetic circuit. Although this naive traversal of the circuit allows for a certain degree of data parallelism, it fails to fully utilize the capacity of modern GPUs. Consequently, many existing frameworks in practice just evaluate the entire computation graph on the CPU (Manhaeve et al., 2018; Ahmed et al., 2022b).

## 3 LAYERIZING CIRCUITS

In this section, we show how to map a d-DNNF circuit to our layerized KLAY representation. Afterward, in Section 4, we discuss how the resulting KLAY representation can be run efficiently on GPUs. Although our exposition focuses on d-DNNF circuits, it is more generally applicable to NNF circuits, meaning KLAY could also be of interest to neurosymbolic systems based on, e.g. fuzzy logic (Badreddine et al., 2022) or grammars (Winters et al., 2022).

### 3.1 FROM LINKED NODES TOWARDS LAYERS

In order to parallelize a d-DNNF circuit, we group the nodes into sets of nodes that can be evaluated in parallel. We dub these groups layers – reminiscent of layers in neural networks. Concretely, for each node $n$ in a circuit $C$ we compute its height in the circuit $h_n$, and nodes with the same height are assigned to the same layer.

$$L_i = \{n \in C \mid h_n = i\} \quad \text{where} \quad h_n = \begin{cases} 0, & \text{if } n \text{ is a leaf,} \\ \max_{c \in \mathcal{C}_n} h_c + 1 & \text{otherwise.} \end{cases} \quad (2)$$

Here, we use $\mathcal{C}_n$ to denote the set of children of a node $n$. Note that the height of all nodes can be efficiently evaluated in a single post-order circuit traversal. The initial layer, $L_0$, comprises all leaf nodes, while the last layer comprises the root node.

Without loss of generality, we can assume that $\vee$-gates only have $\wedge$-gates as children and that $\wedge$-gates have either $\vee$-gates or leaf nodes as children (Choi et al., 2020). This implies that $\wedge$- and $\vee$-gates appear in an alternating fashion throughout the circuit, which can be observed in Figure 1b, and that all nodes in the same layer have the same type.

If nodes are assigned to layers based on their height $h_n$, the child of a node can be in any of the previous layers. However, to transform the circuit evaluation into a sequence of parallel operations, it is more convenient if all children are in the immediately preceding layer. In such a structure, the next layer can be computed solely using the current layer.

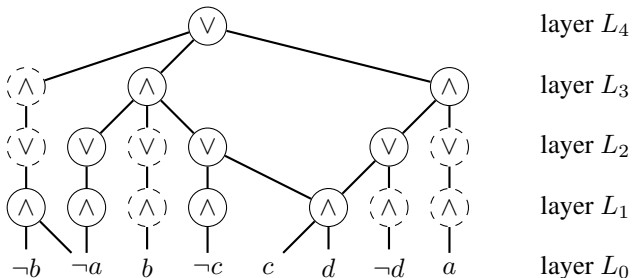

Figure 2: A layered version of the d-DNNF in Figure 1b. Dashed nodes indicate nodes that are not present in the original d-DNNF and which are introduced during the layerization of the circuit.

We obtain this layer-by-layer structure by introducing additional unary nodes. Whenever a node $n \in L_{h_n}$ has a child $c$ in a non-immediately preceding layer $L_{h_c}$, i.e. $h_c + 1 < h_n$, we introduce a chain of unary nodes, one per layer between $L_{h_c}$ and $L_{h_n}$, to connect $n$ to $c$ via these unary nodes. This is illustrated in Figure 2, where the newly introduced nodes are indicated by dashed circles. Note that the type of a unary node ($\vee$ or $\wedge$) is irrelevant and chosen to satisfy the assumption of alternating node types. Algorithm 2 in Appendix C summarizes this layerization in pseudo-code.

### 3.2 MULTI-ROOTED CIRCUITS AND NODE DEDUPLICATION

Suppose now that we have multiple circuits over a (partially) shared set of variables. A set of $M$ circuits effectively computes a Boolean function $f : \mathbb{B}^N \to \mathbb{B}^M$. Such $M$-dimensional circuits are also called multi-rooted circuits and are rather common in the standard neurosymbolic setting. Indeed, unless all data has the same symbolic knowledge, batched training or inference requires multi-rooted circuits. For instance, the widely-known MNIST addition problem (Manhaeve et al., 2018) can be solved using a multi-rooted arithmetic circuit with 19 roots, as it has 19 different possible outputs.

If all $M$ circuits comprising the multi-rooted circuit have already been layerized according to Section 3.1, we again have that nodes at the same height in different circuits are of the same type. We can hence simply combine the layers of the different circuits to further exploit parallelization. Additionally, different circuits might share equivalent sub-circuits amongst each other. By correctly identifying and merging shared nodes, we avoid redundant computations. Figure 3 gives an example of a multi-rooted d-DNNF circuit with shared nodes.

To merge all duplicate nodes, we need an efficient way to identify these nodes. We show that this is possible in linear time.

**Theorem 1.** *Given a set of circuits, all identical sub-circuits can be identified in linear expected time complexity in the total number of edges in the circuits.*

The proof is included in Appendix A. For canonical circuits, such as SDDs with the same variable tree[1], this result is strengthened from syntactically identical sub-circuits to semantic equivalence. We realize Theorem 1 using Merkle hashes (Merkle, 1987). That is, we associate a hash $hash(n)$ with every node $n$ in a recursive fashion. For each leaf node, we hash the associated (negated) variable. For each internal node, we combine the hashes of the children using a permutation invariant function.

$$hash(n) = \begin{cases} \text{mix\_hash}(l_n) & \text{if } n \text{ is leaf.} \\ \bigoplus_{c \in \mathcal{C}_n} \text{mix\_hash}(hash(c)) & \text{otherwise.} \end{cases} \quad (3)$$

Here, $l_n$ is a unique identifier for the (negated) variable of the leaf node $n$, $\bigoplus$ is a permutation-invariant operation such as XOR, and mix\_hash is a function that disperses the bits of the hash.

We can now merge all equivalent nodes in a multi-rooted circuit by computing the node hashes in a bottom-up pass and merging all nodes with the same hash, e.g. by storing the nodes in a hash map.

---

[1] A sentential decision diagram (SDD) is a subclass of d-DNNF circuits that are canonical given a variable tree. We refer to Darwiche (2011) for more details.

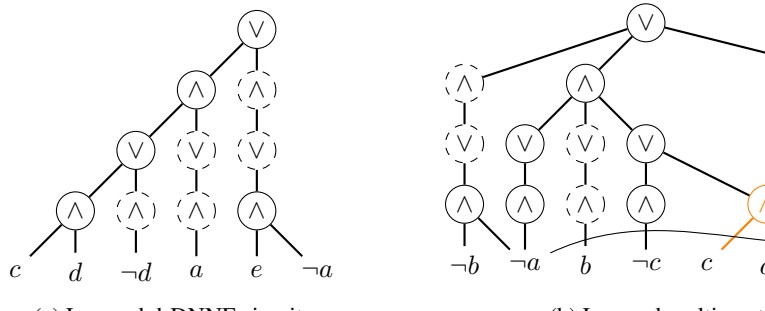

(a) Layered d-DNNF circuit.

(b) Layered multi-rooted circuit.

Figure 3: (a) contains a layered d-DNNF circuit over the variables $a$, $c$, $d$, and $e$. The dashed circles indicate again nodes introduced during the layerization process. (b) illustrates a multi-rooted circuit where the upper left node computes the circuit from Figure 2 and where the upper right node computes the circuit from Figure (a). Note how nodes at the same height in the multi-rooted circuit are of the same type and how the sub-circuit colored in orange is used to compute both root nodes.

The efficient deduplication via Merkle hashes is not only useful in the context of merging single-rooted circuits; duplicate nodes may also occur within a single circuit. This can, for instance, happen when the circuit is an SDD[2]. Another source of duplicate nodes stems from the layerization procedure in Section 3.1. Concretely, the connection of a node $n$ to a higher-up node through a chain of unary nodes results in equivalent chains if node $n$ has multiple higher-up parent nodes. Using Merkle hashes resolves this issue as it automatically merges multiple equivalent node chains, limiting the overhead of adding new nodes.

Lastly, we make sure that all circuits have the same height by artificially extending a circuit with unary nodes at the root if necessary. This results in the last layer of our multi-rooted circuit containing the root nodes of every added circuit.

## 4 TENSORIZING LAYERED CIRCUITS

In the previous section, we organized d-DNNF circuits into layers by assigning each node $n$ in the circuit a height $h_n$. We also avoided duplicate nodes by computing a unique Merkle hash $hash(n)$ for each node. We proceed in this section with mapping the layered linked nodes to a layered computation graph where layers are evaluated sequentially and computations within a layer can be parallelized.

To this end, we make a simple, yet powerful, observation: the current layer can be computed from the output of the previous layer by only using *indexing and aggregation*. To see this, we first impose an arbitrary order on the nodes within each layer. This means we can write the values of all nodes in a layer $L_i$ as a vector $\mathbf{N}_i$. For our running example, we simply pick the order already insinuated by the graphical representation in Figure 2, except for the input layer where we order lexicographically: $\mathbf{N}_0 = [p_a, p_{\neg a}, p_b, p_{\neg b}, p_c, p_{\neg c}, p_d, p_{\neg d}]^\top$.

Now, in order to compute $\mathbf{N}_l$ from $\mathbf{N}_{l-1}$ we use two vectors of indices: $\mathbf{S}_l$ and $\mathbf{R}_l$. For each edge between $\mathbf{N}_{l-1}$ and $\mathbf{N}_l$, $\mathbf{S}_l$ contains the index of the input node, while $\mathbf{R}_l$ contains the index of the output node. We exemplify this for a single layer in Figure 4. By imposing an order on the input nodes and by providing the set of vectors $\{\mathbf{S}_1 \dots \mathbf{S}_L\}$ and $\{\mathbf{R}_1 \dots \mathbf{R}_L\}$, we entirely characterize an arithmetic circuit. This means that we can use these vectors instead of a linked node representation to evaluate the circuit.

To compute $\mathbf{N}_l$, we first select relevant values from $\mathbf{N}_{l-1}$ using as index $\mathbf{S}_l$, giving us $\mathbf{E}_l = \mathbf{N}_{l-1}[\mathbf{S}_l]$. The vector $\mathbf{E}_l$ essentially contains the values of all the edges between $\mathbf{N}_l$ and $\mathbf{N}_{l-1}$. Next, we need to correctly segment the edges $\mathbf{E}_l$ and aggregate the individual segments – either by using sums or products, depending on the layer. This is done using $\mathbf{R}_l$: all elements with the same index in $\mathbf{R}_l$

---

[2]An SDD node represents an $\vee$-gate, and is defined as a set of tuples each representing $\wedge$-gates. This structure typically does not share tuples, and thus does not automatically reuse duplicate $\wedge$-gates.

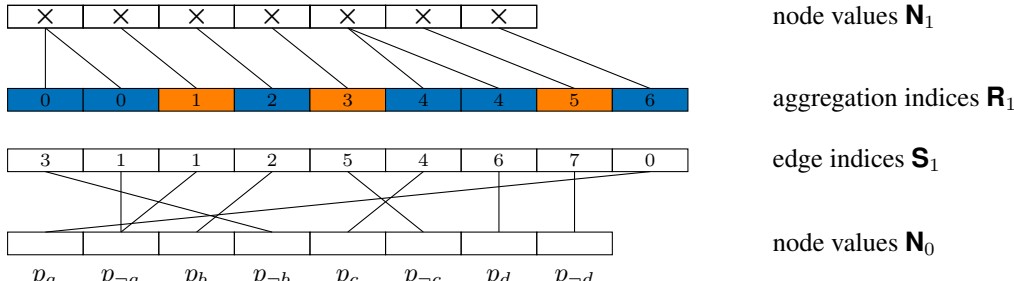

Figure 4: The evaluation of the first layer in Figure 1c, as an indexing and aggregation operation. The symbols $\mathbf{N}_0$ and $\mathbf{N}_1$ denote the node values at layer $L_0$ and $L_1$ respectively. First, we index in $\mathbf{N}_0$ using the edge indices $\mathbf{S}_1$, effectively creating a vector with the values of all edges in between $\mathbf{N}_0$ and $\mathbf{N}_1$. Next, the aggregation indices $\mathbf{R}_1$ determine what edges are reduced together. As a result we obtain the node values $\mathbf{N}_1 = \text{scatter}(\mathbf{N}_0[\mathbf{S}_1], \mathbf{R}_1, \text{reduce} = \text{'product'})$.

are reduced together. Fortunately, such segment-reduce operations are implemented as primitives in various tensor libraries such as Jax, TensorFlow, or PyTorch (Abadi et al., 2015; Paszke et al., 2019; Bradbury et al., 2018). In Figure 4, these segments are indicated with alternating colors.

Algorithm 1 contains pseudo-code for the layerwise circuit evaluations, where we use the common `scatter` function to segment and aggregate the $\mathbf{E}_l$ vectors. In Figure 5, we show the full tensorized circuit representation of the arithmetic circuit in Figure 1c, which corresponds to the computation graph induced by Algorithm 1. Although we depict the edge vectors $\mathbf{E}_l$ in Figure 5, we stress that these do not need to be represented explicitly because modern tensor libraries in practice fuse the indexing and segmentation operations together.

---

**Algorithm 1:** KLay Forward Evaluation

---

**Input:** $\mathbf{N}_0$, selection indices $\mathbf{S}_1, \mathbf{S}_2, \ldots, \mathbf{S}_L$, reduction indices $\mathbf{R}_1, \mathbf{R}_2, \ldots, \mathbf{R}_L$
**for** $l \leftarrow 1$ **to** $L$ **do**
    $\mathbf{E}_l \leftarrow \mathbf{N}_{l-1}[\mathbf{S}_l]$;
    **if** $l \bmod 2 = 0$ **then**
        $\mathbf{N}_l \leftarrow \text{scatter}(\mathbf{E}_l, \mathbf{R}_l, \text{reduce}=\text{'sum'})$;
    **else**
        $\mathbf{N}_l \leftarrow \text{scatter}(\mathbf{E}_l, \mathbf{R}_l, \text{reduce}=\text{'product'})$;
    **end**
**end**
**return** $\mathbf{N}_L$;

---

Finally, we note that in practice arithmetic circuits are usually evaluated in the logarithmic domain for numerical stability. This means that the reduction uses logsumexp and sum operations instead of the sum and product, respectively. We provide the pseudo-code in Algorithm 4 in Appendix C. More generally, any pair of operations that forms a semiring may be used to evaluate the circuit (Kimmig et al., 2017), as long as the semiring operations can be expressed as reduction operations in the tensor library.

## 5 RELATED WORK

The closest related work is the arithmetic circuit layerization present in JUICE (Dang et al., 2021). Similar to our circuit layerization scheme, JUICE takes a Boolean circuit and maps it to a set of layers that can be evaluated sequentially, although not layer per layer. To this end, Dang et al. (2021) implemented a custom SIMD implementation for the CPU and custom CUDA kernels for the GPU. This is in contrast to KLAY where we reduce circuit evaluations to a sequence of index and scatter-reduce operations, which are already efficiently implemented in the modern deep learning stack. In our experiments, we show that KLAY dramatically outperforms JUICE in terms of run time on

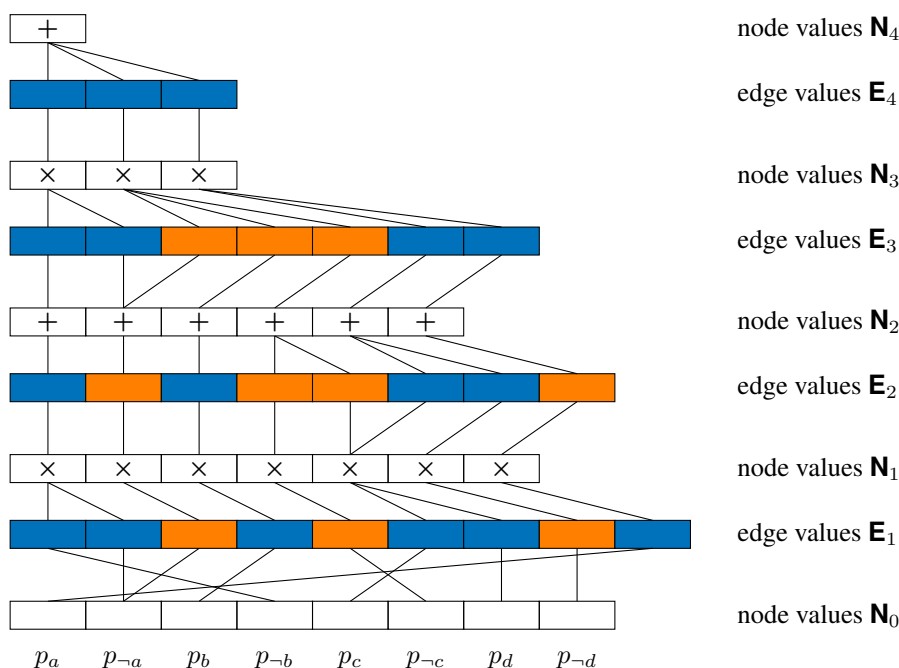

Figure 5: Tensorized KLAY representation of the arithmetic circuit in Figure 1c. We depict the segments of the edge vectors $\mathbf{E}_l$ with alternating colors.

CPU and more importantly on GPU as well. Noteworthy here is that our experimental evaluation also shows that JUICE's GPU implementation is slower than their CPU implementation – hinting at missed parallelization opportunities.

The difficulty of running arithmetic circuits on GPUs was also pointed out by Shah et al. (2020; 2021). While Shah et al. (2020; 2021) focused on developing hardware accelerators for arithmetic circuits, they also implemented custom circuit evaluations exploiting to a certain degree SIMD instructions and GPU parallelization. They found that CPU circuit evaluations outperformed the GPU implementations. Shah et al. (2020) concluded that arithmetic circuits were simply too sparse to be run efficiently on GPUs. By layerizing arithmetic circuits and interpreting the product and sum layers as indexing and scatter-reduce operations, we are able to refute this claim. We provide experimental evidence for this in Section 6.

Notable is also the work of Vasimuddin et al. (2018), who proposed an efficient CPU implementation for evaluating arithmetic circuits deployed on multiple CPU cores. They deemed an efficient GPU implementation to be impractical due to the high demands on memory bandwidth. We can again refute this claim.

Arithmetic circuits are closely related to the model class of probabilistic circuits (Vergari et al., 2021). The main difference is that the sum units for the latter are parameterized using mixture weights, while the former does not contain such weights. In order to run probabilistic circuits efficiently on the GPU, implementations usually rely on casting circuit evaluations as dense matrix-vector product (Peharz et al., 2020b;a; Galindez Olascoaga et al., 2019; Mari et al., 2023; Sommer et al., 2021). This idea has recently been generalized by Liu et al. (2024), who allow for the presence of block-sparsity in the matrix that encodes a layer. By exploiting this sparsity via custom kernels, they widened the class of probabilistic circuits that can be run efficiently on GPUs. Nevertheless, probabilistic circuits are usually far more dense than arithmetic circuits that are compiled from a logical theory. Unfortunately, this prevents the techniques developed for probabilistic circuits to be effective in the context of arithmetic circuits. Similarly, having densely parameterized sum nodes also limits the relevance of techniques developed for arithmetic circuits for probabilistic circuits, e.g. the indexing and segmenting scheme of KLAY circuit evaluations.

Besides algorithmic advances, specialized hardware solutions have also been proposed to deal with the irregularity of the computational graph of an arithmetic circuit (Dadu et al., 2019; Shah et al., 2020; 2021). However, these approaches have the drawback that they would require the purchase of non-commoditized hardware. While this could partially be remedied by the use of FPGAs, as done by Sommer et al. (2018); Weber et al. (2022); Choi et al. (2023), any custom hardware retains a communication overhead. Specifically, in the context of neurosymbolic AI, one needs to pass the output of a neural network to an arithmetic circuit. If the neural net and the circuit are on two different devices, e.g. a GPU and an FPGA, the latency of data transfer can counteract gains in evaluation speed.

# 6    EXPERIMENTAL EVALUATION

We implement KLAY as a Python library supporting two popular tensor libraries: PyTorch and Jax. We evaluate the runtime performance of KLAY on several synthetic benchmarks and neurosymbolic experiments. All experiments were conducted on the same machine, which has an NVIDIA GeForce RTX 4090 as GPU and an Intel i9-13900K as CPU.

## 6.1    SYNTHETIC BENCHMARKS

We first consider the performance of KLAY on a set of synthetic circuits, by randomly generating logical formulas in 3-CNF. We compile the 3-CNF formulas into d-DNNF circuits, more specifically SDD circuits, using the PySDD library[3]. By changing the number of variables and clauses in the CNF, we vary the size of the compiled circuits over 5 orders of magnitude. Figure 6 compares the performance of KLAY with the native post-order traversal from PySDD implemented in C. We report results for both the real and logarithmic semiring. As JUICE does not support the logarithmic semiring and Jax does not support backpropagation on scatter multiplication, these are excluded from the respective comparisons. In Appendix B, we repeat the same experiment using the D4 knowledge compiler (Lagniez & Marquis, 2017) instead of PySDD.

Our results in Figure 6 indicate that on large circuits, KLAY on GPU outperforms all baselines with over one order of magnitude. Due to SIMD and multi-core parallelization, KLAY on CPU is still considerably faster than the baselines. JUICE does not include results for the largest instances due to a timeout after 30 minutes. KLAY attains best results with Jax, due to its superior JIT compilation and kernel fusion.

In Figure 7, we display the size and sparsity of the circuits in our synthetic benchmark. Remarkably, KLAY on average has fewer nodes than the original SDD, meaning the node deduplication outweighs the overhead of introducing unary node chains. As we discuss in Appendix B, this is not the case for circuits compiled by D4. Figure 7 (right) shows that larger circuits are increasingly sparse, with less than one in a million edges being present for the largest circuits.

## 6.2    NEUROSYMBOLIC BENCHMARKS

Next, we consider the use of KLAY in neurosymbolic learning, by measuring the runtime of calculating the gradient on the circuits of several neurosymbolic tasks. As a baseline, we consider the naive evaluation in PyTorch, which is the standard approach performed by existing exact probabilistic neurosymbolic approaches where each node is evaluated individually. We take four different neurosymbolic benchmarks from the literature. The Sudoku benchmark is a classification problem, determining whether a $4 \times 4$ grid of images forms a valid Sudoku (Augustine et al., 2022). The Grid (Xu et al., 2018) and Warcraft (Pogančić et al., 2020) instances require the prediction of a valid low-cost path. Finally, hierarchical multi-level classification (HMLC) concerns the consistent classification of labels in a hierarchy (Giunchiglia & Lukasiewicz, 2020). The results in Table 1 demonstrate the drastic speedups of KLAY compared to the naive baseline, improving by as far as four orders of magnitude for the Warcraft experiment.

As a last experiment, we demonstrate the integration of KLAY into a neurosymbolic framework. Specifically, we use KLAY instead of conventional SDD circuits in DeepProbLog. We measure the

---

[3]https://github.com/wannesm/PySDD

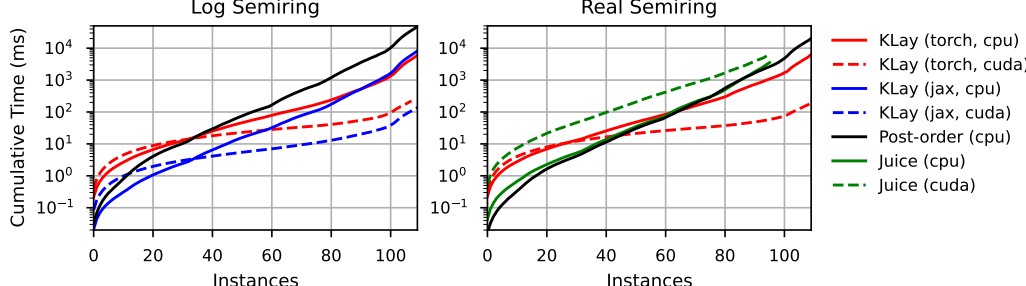

Figure 6: Cumulative runtime in milliseconds for the forward and backward pass on a circuit. That is, we plot the combined run time of the $n$ fastest circuit evaluations against the number of $n$ circuits. Timings are averaged over 10 runs per SDD. The SDDs are randomly generated from 3-CNF, where the difficulty is varied by the number of variables and clauses. The left and right figure show cumulative evaluation times for the logarithmic and real semiring, respectively.

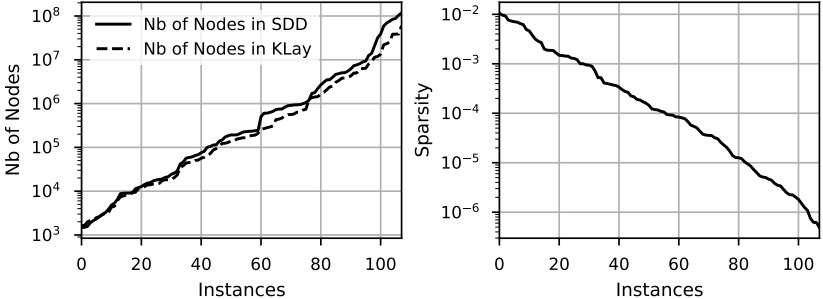

Figure 7: (Left) Number of nodes in the original SDD, compared to the KLAY representation. The instances are the same as in Figure 6. (Right) The sparsity of KLAY, meaning the number of edges in the circuit divided by the number of edges in a dense interconnection of all layers.

Table 1: (Top) Number of nodes in the SDD circuits of various neurosymbolic benchmarks, and the number of nodes and layers in the corresponding KLAY representation. (Middle) Runtime in milliseconds for one forward and backward pass on the circuit with batch size 128, using PyTorch autograd for both KLAY and the naive baseline. (Bottom) Runtime in milliseconds for one forward and backward pass on the neural network of the neurosymbolic task.

|  | Sudoku ($4\times4$) | Grid ($4\times4$) | HMLC | Warcraft ($12\times12$) |
|---|---|---|---|---|
| SDD Nodes | 2 408 | 6 610 | 9 730 | 1 110 234 |
| KLAY Nodes | 3 926 | 6 590 | 26 306 | 1 155 063 |
| KLAY Layers | 30 | 24 | 84 | 84 |
| Naive (cpu) | $17.35 \pm 0.05$ | $46.12 \pm 0.49$ | $127.00 \pm 140.03$ | $11484.53 \pm 251.94$ |
| Naive (cuda) | $42.70 \pm 2.76$ | $107.78 \pm 4.67$ | $175.58 \pm\ \ 1.26$ | $18930.67 \pm 496.16$ |
| KLAY (cpu) | $0.45 \pm 0.02$ | $0.53 \pm 0.03$ | $1.41 \pm\ \ 0.02$ | $18.55 \pm\ \ 0.14$ |
| KLAY (cuda) | $0.46 \pm 0.17$ | $0.49 \pm 0.19$ | $1.03 \pm\ \ 0.35$ | $1.59 \pm\ \ 0.25$ |
| Neural (cpu) | $0.90 \pm 0.63$ | $0.17 \pm 0.02$ | $0.55 \pm\ \ 0.29$ | $242.60 \pm\ \ 4.81$ |
| Neural (cuda) | $0.51 \pm 0.13$ | $0.34 \pm 0.12$ | $0.20 \pm\ \ 0.04$ | $2.80 \pm\ \ 0.07$ |

Table 2: MNIST-addition training time in seconds for one epoch. We report the average and standard deviation over 10 epochs, using a batch size of 128. Both KLAY and the baselines use PyTorch as autograd. T/O indicates a timeout after 2 hours.

| Nb of Digits | 1 | 2 | 3 |
|---|---|---|---|
| KLAY Nodes | 776 | 12 660 | 1 015 867 |
| DeepProbLog (cpu) | $241.63 \pm 4.38$ | $323.26 \pm 22.20$ | T/O |
| Scallop (cpu, $k = 3$) | $52.30 \pm 0.37$ | $601.20 \pm 14.65$ | $5345.78 \pm 171.47$ |
| KLAY (cpu) | $2.08 \pm 0.12$ | $2.10 \pm 0.14$ | $71.54 \pm 0.46$ |
| KLAY (cuda) | $1.26 \pm 0.03$ | $1.25 \pm 0.01$ | $8.85 \pm 0.02$ |

training time of the MNIST-addition task in Table 2. MNIST-addition is a common neurosymbolic task where the input is two numbers represented as MNIST images and the model needs to predict their sum. For more details, we refer to Manhaeve et al. (2018). As a reference, we also include Scallop, which aims to improve the scalability of DeepProbLog by approximating using top-k provenance (Li et al., 2023). While DeepProbLog and Scallop cannot perform batched inference, KLAY can by using multi-rooted circuits. This is reflected in speed-ups of two orders of magnitude over the existing DeepProbLog implementation, even on rather small circuits. Even though KLAY remains exact unlike Scallop, KLAY also demonstrates large speedups here.

## 7    CONCLUSIONS

The success of neural networks has been largely attributed to their scale (Kaplan et al., 2020), which is realized by their effective use of hardware accelerators. To compete, novel methods must run efficiently on the available hardware or risk losing out due to what Hooker (2021) coined the hardware lottery. We tackled this issue for neurosymbolic AI by introducing KLAY– a new data structure to represent arithmetic circuits that is amenable to efficient evaluations on modern GPUs or TPUs. Along with this representation, we contributed three algorithms for KLAY. The first two algorithms map the traditional linked node representation of arithmetic circuits to the corresponding KLAY representation (Algorithm 2 and 3). This representation is efficiently evaluated using the third algorithm, which exploits the parallelization opportunities (Algorithm 1).

Contrary to widely held belief, our experiments demonstrated that arithmetic circuits can be run efficiently on GPUs, despite their high degree of sparsity. To this end, a key aspect is the reduction of circuit evaluations to a sequence of indexing and scatter-reduce operations, as these can be implemented using highly optimized primitives available in modern tensor libraries. Resolving circuit evaluations as one of the major bottlenecks present in current neurosymbolic architectures allows to further scale neurosymbolic models and tackle new problems that have been out of reach so far.

AUTHOR CONTRIBUTIONS

PZD conceived the conceptual idea of KLAY and made an initial prototype. JM worked out the technical details and implemented the methods with the help of VD. JM proposed and carried out the experiments. All authors contributed equally to the writing. PZD supervised the project.

ACKNOWLEDGMENTS

This research received funding from the Flemish Government (AI Research Program), the Flanders Research Foundation (FWO) under project G097720N, the KU Leuven Research Fund (C14/18/062), and the European Research Council (ERC) under the European Union's Horizon 2020 research and innovation programme (Grant agreement No. 101142702). PZD is supported by the Wallenberg AI, Autonomous Systems and Software Program (WASP) funded by the Knut and Alice Wallenberg Foundation. We thank Luc De Raedt for his valuable feedback.

REPRODUCIBILITY

The functionality of KLAY has been described in pseudo-code (Algorithms 1, 2, and 3) and has been implemented as a Python library to easily replicate all experiments in the paper, available at `https://github.com/ML-KULeuven/klay`.

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

# APPENDIX

## A  PROOF OF THEOREM 1

**Theorem 1.** *Given a set of circuits, all identical sub-circuits can be identified in linear expected time complexity in the total number of edges in the circuits.*

*Proof.* We provide a constructive proof. First, observe that nodes representing syntactically identical sub-circuits have the same height and are therefore members of the same layer. Second, observe that nodes within a layer are identified by their set of children. Combining these two observations, we can rely on the following inductive property: by ensuring deduplication of all nodes in layer $L_{n-1}$, it is trivial to check equivalence for the nodes in layer $L_n$. In conclusion, we can identify and deduplicate all equivalent nodes by iterating over all nodes bottom-up. To detect equivalent nodes within a layer, we employ an efficient hashing scheme where each node $n$ is hashed as the set of the hashes of its children $\mathcal{C}_n$ (see Equation 3). As the set of nodes is fixed, we can assume a perfect hashing scheme devoid of any hashing conflicts (Belazzougui et al., 2009). As this algorithm considers each edge precisely once, it is indeed linear in the number of edges. Moreover, the hash tables requires linear memory in the number of nodes. □

## B  ADDITIONAL EXPERIMENTAL DETAILS

**Synthetic Benchmarks**  To generate the circuits, we randomly generate 3-CNF formulas. This requires 2 parameters, the number of clauses $k$ and the number of variables $v$. The generated 3-CNF formulas contains $k$ clauses with 3 randomly picked variables from the $v$ variables. To vary the difficulty, we generate 3-CNF with 30, 35, 40, 45, 50, 55, 60, 65, 70, 75, and 80 variables. For each variable count, we generate 10 random instances, resulting in a total of 110 instances. The clause count is also taken as half the number of variables for the SDD benchmarks and twice the number of variables for the D4 benchmarks. This difference can be attributed to D4 generating smaller circuits, so we make the instances harder.

**Extra Experiments**  In Figure 8, we repeat the same synthetic experiment as in Figure 6 but using a top-down d-DNNF knowledge compiler, D4, instead of a bottom-up SDD compiler. As a baseline, we use an optimized Rust implementation of the node-by-node post-order evaluation algorithm. These circuits do not contain duplicate nodes and are somewhat less balanced, leading to a larger overhead in terms of extra nodes, as is visible on the right of Figure 8. Nonetheless, KLAY still outperforms the baseline by a large margin. Finally, in Table 1 we repeat the neurosymbolic experiments from Table 1 with batch size 128 instead of 1.

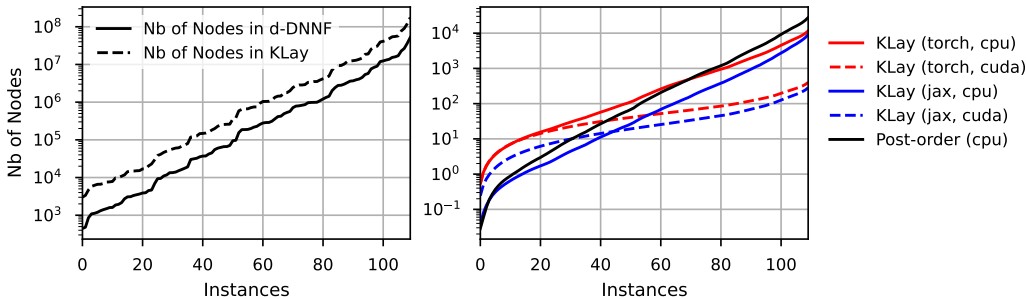

Figure 8: (Left) Number of nodes in the original d-DNNF, compared to KLAY's layerized circuit. (Right) Cumulative runtime in milliseconds for the forward and backward pass on a d-DNNF circuit in the log semiring. Timings for each individual circuit are averaged over 10 runs. Each instance is a randomly generated logical formula in 3-CNF, compiled into a d-DNNF circuit using D4. The number of variables and clauses in the CNF was varied to achieve different levels of difficulty.

Table 3: (Top) Number of nodes in the SDD circuits of various neurosymbolic benchmarks, and the number of nodes and layers in the corresponding KLAY representation. (Middle) Runtime in milliseconds for one forward and backward pass on the circuit with batch size 128, using PyTorch autograd for both KLAY and the naive baseline. (Bottom) Runtime in milliseconds for one forward and backward pass on the neural network of the neurosymbolic task.

|  | Sudoku (4×4) | Path (4×4) | HMLC | Warcraft (12×12) |
|---|---|---|---|---|
| SDD Nodes | 2 408 | 6 610 | 9 730 | 1 110 234 |
| KLAY Nodes | 3 926 | 6 590 | 26 306 | 1 155 063 |
| KLAY Layers | 30 | 24 | 84 | 84 |
| Naive (cpu) | $21.55 \pm 0.83$ | $55.46 \pm 5.83$ | $122.37 \pm 35.33$ | $12225.93 \pm 626.03$ |
| Naive (cuda) | $47.20 \pm 5.29$ | $112.86 \pm 2.49$ | $201.07 \pm 18.62$ | $19324.75 \pm 74.16$ |
| KLAY (cpu) | $5.64 \pm 0.06$ | $4.96 \pm 0.15$ | $15.43 \pm 0.17$ | $514.20 \pm 22.14$ |
| KLAY (cuda) | $3.03 \pm 0.05$ | $2.47 \pm 0.03$ | $8.76 \pm 1.56$ | $72.59 \pm 0.16$ |
| Neural (cpu) | $23.05 \pm 3.63$ | $0.72 \pm 0.02$ | $0.63 \pm 0.15$ | $17420.27 \pm 202.31$ |
| Neural (cuda) | $1.24 \pm 0.21$ | $0.30 \pm 0.02$ | $0.21 \pm 0.04$ | $73.20 \pm 0.02$ |

## C PSEUDO-CODE

Algorithm 2 and 3 contains pseudo-code of the previously discussed layerization and tensorization procedures. Algorithm 4 contains the pseudo-code for the evaluation of KLAY in the logarithmic semiring instead of the real semiring.

---

**Algorithm 2:** KLay Layerization

---

**Input:** Boolean Circuit $C$ as linked nodes.

layers $\leftarrow [\,]$;
height $\leftarrow [\,]$;
hashes $\leftarrow [\,]$;
**for** *node $n$ in the nodes of $C$, children before parents* **do**
  **if** $n$ *is a leaf node* **then**
    height$[n] \leftarrow 0$ ;
    hashes$[n] \leftarrow hash(n)$;
  **else**
    height$[n] \leftarrow 1 + \max_{c \in \text{children}(n)} \text{height}[c]$;
    /* Bring children to the prior layer */
    **for** *child node $c$ in children(n)* **do**
      **while** *height[c]+1 $\neq$ height[n]* **do**
        $c \leftarrow$ new unary node with child $c$;
        height$[c] \leftarrow \text{height}[\text{child}(c)] + 1$;
        hashes$[c] \leftarrow hash(\text{hashes}[\text{child}(c)])$;
        layers$[\text{height}[c]][\text{hashes}[c]] \leftarrow c$;
      **end**
    **end**
    hashes$[n] \leftarrow \bigoplus_{c \in \text{children}(n)} hash(\text{hashes}[c])$
  **end**
  /* Add node to its layer */
  layers$[\text{height}[n]][\text{hashes}[n]] \leftarrow n$;
**end**
**return** *layers*;

---

---

**Algorithm 3:** KLay Tensorization

---

**Input:** layers.

/* We assume the nodes in each layer are ordered, such that index(n) is the index of node n in its layer. */

**for** *layer l in layers* **do**

    $\mathbf{S}_l \leftarrow [\,]$;

    $\mathbf{R}_l \leftarrow [\,]$;

    **for** *node n in layer l* **do**

        **for** *child c of node n* **do**

            $\mathbf{R}_l$.push(index($n$));

            $\mathbf{S}_l$.push(index($c$));

        **end**

    **end**

**end**

**return** $\{\mathbf{S}_1, \ldots, \mathbf{S}_L\}, \{\mathbf{R}_1, \ldots, \mathbf{R}_L\}$;

---

**Algorithm 4:** KLay Forward Evaluation in the Logarithmic Semiring

---

**Input:** $\mathbf{N}_0$, selection indices $\mathbf{S}_1, \mathbf{S}_2, \ldots, \mathbf{S}_L$, reduction indices $\mathbf{R}_1, \mathbf{R}_2, \ldots, \mathbf{R}_L$, epsilon $\epsilon$

**for** $l \leftarrow 1$ **to** $L$ **do**

    $\mathbf{E}_l \leftarrow \mathbf{N}_{l-1}[\mathbf{S}_l]$;

    **if** $l \bmod 2 = 0$ **then**

        /* As logsumexp reduction is not always implemented for scatter, we perform it using max and sum scatter operations. */

        $\mathbf{M} \leftarrow \text{scatter}(\mathbf{E}_l, \mathbf{R}_l, \text{reduce=`max'})$;

        $\mathbf{T} \leftarrow \text{scatter}(\exp(\mathbf{E}_l - \mathbf{M}[\mathbf{R}_l]), \mathbf{R}_l, \text{reduce=`sum'})$;

        $\mathbf{N}_l \leftarrow \log(\mathbf{T} + \epsilon) + \mathbf{M}$;

    **else**

        $\mathbf{N}_l \leftarrow \text{scatter}(\mathbf{E}_l, \mathbf{R}_l, \text{reduce=`sum'})$;

    **end**

**end**

**return** $\mathbf{N}_L$;

---

## D  NEUROSYMBOLIC EXAMPLE

We demonstrate how neural networks can be combined with circuits on a simplified variant of the canonical MNIST-addition task (Manhaeve et al., 2018). In this task, two MNIST images are given containing either the digit zero or one, and the goal is to predict the sum of the two digits. Note that the labels of the individual digits are not given. There are four distinct possibilities for the ground truth labels of the images. Either both images are zero and the sum is also zero, or one image is zero and the other is one resulting in a sum of one, or both are one and the sum is two.

Let us write $p(\text{Img}_i = j)$ for the probability that image $i$ contains the digit $j$, and $p(\text{Sum} = k)$ for the probability that the sum equals $k$. Now it follows that

$$p(\text{Sum} = 0) = p(\text{Img}_0 = 0) \cdot p(\text{Img}_1 = 0)$$
$$p(\text{Sum} = 1) = p(\text{Img}_0 = 0) \cdot p(\text{Img}_1 = 1) + p(\text{Img}_0 = 1) \cdot p(\text{Img}_1 = 0)$$
$$p(\text{Sum} = 2) = p(\text{Img}_0 = 1) \cdot p(\text{Img}_1 = 1)$$

We can now encode the above probabilities in a multi-rooted circuit, which we visualize in Figure 9. Observe that each root computes one of the probabilities for $p(\text{Sum} = k)$.

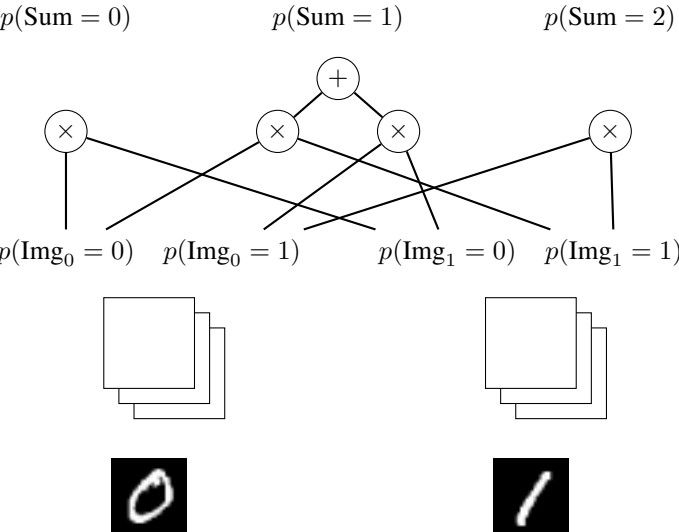

Figure 9: A simple neurosymbolic model for the MNIST-addition task. Two MNIST images are given as input. First, a vision model classifies the probability of the image containing a zero or one. Next, a circuit uses these probabilities to calculate the probabilities of the different possible sums.

