# OpenReview forum: "KLay: Accelerating Arithmetic Circuits for Neurosymbolic AI"
_ICLR.cc/2025/Conference — ICLR 2025 Poster_

### Official Review · Reviewer_Jxpy · 2024-10-25

**Soundness:** 4
**Presentation:** 3
**Contribution:** 3
**Rating:** 6
**Confidence:** 3

**Summary:**

This paper proposes a method to tackle a numerical problem in the field of neurosymbolic AI. It allows to represent arithmetic circuits, used to model symbolic constraints, as a new kind of data structure, knowledge layers (KLay), on which computations can be performed more efficiently. Notably, computations on KLay can be performed in parallel, and thus exploit the full potential of AI accelerators such as GPUs. Empirical results show the superiority of these layers over existing methods in terms of computational speed.

**Strengths:**

+ The paper tackles an apparently difficult problem, which other authors had claimed is impossible to solve
+ It paves the way for a broader deployment of neurosymbolic AI models on standard deep learning hardwares such as GPU or TPU, while efficiently exploiting their potential
+ Empirical results are displayed in a clear way, showing the effective superiority of the proposed method over existing ones

**Weaknesses:**

No weekness to mention

**Questions:**

How are the KLay optimized ? Is optimization performed on the arithmetic circuits (that are emphasized as differentiable in the text) before representing them as KLay ? Could the authors address this question more explicitly would benefit to the completeness of the analysis.

---

> ### Author Response · Authors · 2024-11-17
> **Rebuttal**
>
> > How are the KLay optimized ?
>
> We optimize the parameters of a circuit with conventional gradient-based optimizers, specifically, Adam (line 125). Note also that we do not learn the structure of the circuit, as this structure is generated by a knowledge compiler such as PySDD (lines 86-93).
>
> > Is optimization performed on the arithmetic circuits (that are emphasized as differentiable in the text) before representing them as KLay ?
>
> Both inference and training using KLay is considerably faster (see e.g. Table 1). Hence, the point is to first convert the circuit to KLay before training.

---

> > ### Comment · Reviewer_Jxpy · 2024-11-29
> > **Reply to the authors**
> >
> > Thanks to the authors for addressing my questions. It does not change my initial overall appreciation on the paper, and i will maintain my score of 6.

---

### Official Review · Reviewer_EziM · 2024-10-27

**Soundness:** 3
**Presentation:** 3
**Contribution:** 3
**Rating:** 6
**Confidence:** 4

**Summary:**

The authors propose KLay, a data structure that enable parallel computing on irregular operations in knowledge layers of neurosymbolic AI.

**Strengths:**

The problem the authors formulated is clear.

**Weaknesses:**

The motivation needs further justification.

**Questions:**

I appreciate the authors effort in advancing neurosymbolic AI, and I personally am educated a lot from this work. Below are some questions, mainly regarding the motivation.

1. "arithmetic circuits are ill-suited to be evaluated on AI accelerators..." Additional to GPUs/TPUs, AI accelerators also use ASICs and FPGAs as platforms, which are known to be capable of handling irregular operations. It would be insightful if the authors could discuss why arithmetic circuits are not suitable on these platforms.

2. Could you please clarify if the neural networks are indeed separate from the arithmetic circuit? My understanding is that the leaf node values represent probabilities obtained from neural networks, and the subsequent operations involve arithmetic computations from the leaf nodes to the root node. Would it be practical to transfer the probability data to CPUs for post-order tree traversal after the neural network computations are performed on GPUs?

3. Could the authors provide insights into how the workload and time consumption of the arithmetic circuit compare to the upstream neural networks? Is there a typical percentage that represents this comparison?

4. Could the authors clarify if this approach can be considered an efficient method for parallel post-order tree traversal? If so, how does it compare to other efficient parallel algorithms for post-order tree traversal, if any exist?

5. Do current AI models possess the capabilities that neurosymbolic AI offers? What are the advantages compared to modern AI models do the authors anticipate that neurosymbolic AI can achieve? It would be very helpful if examples can be provided.

---

> ### Author Response · Authors · 2024-11-17
> **Rebuttal (part 1)**
>
> We are glad to read that the problem was clearly formulated, that there are no weaknesses barring the motivation, and that the work has personally educated you. In this light, we were surprised by the low score. We have added a section in the Appendix to motivate the use of circuits with neural networks, and we hope to address your remaining questions below.
>
> > "arithmetic circuits are ill-suited to be evaluated on AI accelerators..." Additional to GPUs/TPUs, AI accelerators also use ASICs and FPGAs as platforms, which are known to be capable of handling irregular operations. It would be insightful if the authors could discuss why arithmetic circuits are not suitable on these platforms.
>
> Efficiently executing circuits on FPGAs is certainly possible [1, 2]. However, as pointed out in the paper (lines 380-388) it remains desirable that neurosymbolic methods can be trained on the same device as the neural network, which in practice usually means either GPUs or TPUs. Not only are FGPAs much less widely available to practitioners (line 383), they also induce considerable latency as the probabilities and gradients need to be transferred between GPU and FPGA every iteration (line 386).
>
> [1]: Sommer, Lukas, et al. "Automatic mapping of the sum-product network inference problem to fpga-based accelerators." 2018 IEEE 36th International Conference on Computer Design (ICCD). IEEE, 2018.
>
> [2]: Choi, Young-kyu, et al. "FPGA acceleration of probabilistic sentential decision diagrams with high-level synthesis." ACM Transactions on Reconfigurable Technology and Systems 16.2 (2023): 1-22.
>
> > Could you please clarify if the neural networks are indeed separate from the arithmetic circuit? My understanding is that the leaf node values represent probabilities obtained from neural networks, and the subsequent operations involve arithmetic computations from the leaf nodes to the root node.
>
> Yes, this is correct (see lines 107-125). This is an established technique in neurosymbolic AI [1,2,3,4].
>
> [1]: Xu, Jingyi, et al. "A semantic loss function for deep learning with symbolic knowledge." International conference on machine learning (2018).
>
> [2]: Manhaeve, Robin, et al. "Deepproblog: Neural probabilistic logic programming." Advances in neural information processing systems 31 (2018).
>
> [3]: Maene, Jaron, et al. "On the Hardness of Probabilistic Neurosymbolic Learning." Forty-first International Conference on Machine Learning (2024).
>
> [4]: Ahmed, Kareem, et al. "Semantic probabilistic layers for neuro-symbolic learning." Advances in Neural Information Processing Systems 35 (2022).
>
> > Would it be practical to transfer the probability data to CPUs for post-order tree traversal after the neural network computations are performed on GPUs?
>
> No, this is not practical. Firstly, the communication between CPU and GPU introduces considerable latency. Secondly, post-order evaluation on CPU is not parallel and hence many orders of magnitudes slower compared to leveraging GPUs (see Figure 6). Note that even on CPU our method is still 10x faster than post-order traversal (see Figure 6).
>
> > Could the authors clarify if this approach can be considered an efficient method for parallel post-order tree traversal? If so, how does it compare to other efficient parallel algorithms for post-order tree traversal, if any exist?
>
> There do exist methods for parallel tree traversal on the GPU. However, they are not applicable in our setting as 1) we have a DAG and not a tree and 2) we have alternating node types inside of the DAG. In this regard, our method is not commensurable with existing GPU tree traversal methods.

---

> ### Author Response · Authors · 2024-11-17
> **Rebuttal (part 2)**
>
> > Could the authors provide insights into how the workload and time consumption of the arithmetic circuit compare to the upstream neural networks? Is there a typical percentage that represents this comparison?
>
> The circuit evaluation time typically dominates the inference in neurosymbolic experiments. For instance, when running the MNIST-addition experiment with SoTA methods (e.g. Scallop or DeepProbLog), the neural network takes less than 1% of the inference runtime. For reference, we have added the neural network inference timings for the considered neurosymbolic tasks in Table 1 and 3. Although the precise ratio between circuits and neural nets depends on the specific experiment setup, the absolute improvement in execution time of KLay is very significant as long as the circuit is large (see Figure 6), regardless of the choice of neural net.
>
> > Do current AI models possess the capabilities that neurosymbolic AI offers? What are the advantages compared to modern AI models do the authors anticipate that neurosymbolic AI can achieve? It would be very helpful if examples can be provided.
>
> As mentioned in the first paragraph, “including symbolic information has been shown to increase reasoning capabilities [1, 2], safety [3], controllability [4], and interpretability [8]. Furthermore, neurosymbolic methods often require less data by allowing a richer and more explicit set of priors [5, 6].”
>
> As a concrete example, neurosymbolic methods can generate code that is guaranteed to be executable [4], or generate predictions for self-driving cars that are always consistent with common-sense constraints [7]. On the other hand, it is impossible for pure neural approaches to attain similar guarantees.
>
> [1]: Yi, Kexin, et al. "Neural-symbolic vqa: Disentangling reasoning from vision and language understanding." Advances in neural information processing systems 31 (2018).
>
> [2]: Trinh, Trieu H., et al. "Solving olympiad geometry without human demonstrations." Nature 625.7995 (2024).
>
> [3]: Yang, Wen-Chi, et al. "Safe reinforcement learning via probabilistic logic shields." Proceedings of the Thirty-Second International Joint Conference on Artificial Intelligence. 2023.
>
> [4]: Jiao, Ying, et al. "Valid text-to-sql generation with unification-based deepstochlog." International Conference on Neural-Symbolic Learning and Reasoning, 2024.
>
> [5]: Diligenti, Michelangelo, Marco Gori, and Claudio Sacca. "Semantic-based regularization for learning and inference." Artificial Intelligence 244 (2017): 143-165.
>
> [6]: Manhaeve, Robin, et al. "Deepproblog: Neural probabilistic logic programming." Advances in neural information processing systems 31 (2018).
>
> [7]: Giunchiglia, Eleonora, et al. "ROAD-R: the autonomous driving dataset with logical requirements." Machine Learning 112.9 (2023).
>
> [8]: Koh, Pang Wei, et al. "Concept bottleneck models." International conference on machine learning. 2020.

---

> > ### Comment · Reviewer_EziM · 2024-11-17
> > **Follow-up discussions**
> >
> > Thank you for your clarification. However, there are several minor points that not convincing enough to me. In my humble opinion, the importance of motivation is sometimes above everything else.
> >
> > 1. I respectfully disagree with the authors' definition of AI accelerators, as dataflow-architecture-based AI accelerators is a well-established research domain. To list a few:
> > [1]: Sixu Li, et al. "Fusion-3D: Integrated Acceleration for Instant 3D Reconstruction and Real-Time Rendering." (MICRO'24, best paper award)
> > [2]: Dongseok Im, et al. "LUTein: Dense-Sparse Bit-slice Architecture with Radix-4 LUT-based Slice-Tensor Processing Units." (HPCA'24)
> > [3]: Yonggan Fu, et al. "Gen-NeRF: Efficient and Generalizable Neural Radiance Fields via Algorithm-Hardware Co-Design." (ISCA '23).
> > [4]: Cong Guo, et al. "ANT: Exploiting Adaptive Numerical Data Type for Low-bit Deep Neural Network Quantization." (MICRO'22)
> >
> > 2. FPGAs are not as widely applicable as GPUs - the point is well taken. However, the expensive data transfer between FPGA and GPU is not quite clear to me, as training and inference can be theoretically implemented on the same FPGA - the benchmarks seem small enough to be put on a single FPGA, since modern FPGAs already have considerable capacity. Could the authors explain if this is not the case? Are there deeper reasons that data transfer is necessary?
> >
> > Some follow-up questions:
> > 1. As far as I understand, it is implied that parallel DAG traversal on GPU is a much greater challenge than parallel tree traversal. It would be helpful if you can confirm if this is true and elaborate.
> >
> > 2. The authors made a good point of controllability and interpretability. Out of curiosity, do the authors think if applying this technique on, for example, Large Language Models, can help solving the hallucination problem?
> >
> > Thank you again for your contributions, and I will increase the score once everything makes sense to me.

---

> > > ### Author Response · Authors · 2024-11-19
> > >
> > > > I respectfully disagree with the authors' definition of AI accelerators, as dataflow-architecture-based AI accelerators is a well-established research domain. To list a few: (...)
> > >
> > > We agree that our use of the term AI accelerators may have been imprecise and did not wish to exclude alternative forms of hardware. We have removed the term in our paper to avoid confusion and now only make reference to GPUs and TPUS.
> > >
> > > > FPGAs are not as widely applicable as GPUs - the point is well taken. However, the expensive data transfer between FPGA and GPU is not quite clear to me, as training and inference can be theoretically implemented on the same FPGA - the benchmarks seem small enough to be put on a single FPGA, since modern FPGAs already have considerable capacity. Could the authors explain if this is not the case? Are there deeper reasons that data transfer is necessary?
> > >
> > > To make sure that the motivation of our work is completely clear, we go into more detail of why KLay was developed and how it relates to the concerns about FPGAs.
> > >
> > > 1) The primary goal of KLay is to make efficient neurosymbolic AI more accessible.
> > >
> > > 2) As training neural networks on GPUs with PyTorch/Tensorflow/Jax is de facto standard in the research community, it would hence be desirable this same toolbox could be effectively harnessed for neurosymbolic AI. Although there are plenty of existing works that accelerate circuits on FPGA or even custom hardware, practitioners typically do not have access to these. Hence, all FPGA solutions have seen little adoption in the neurosymbolic community and the de facto standard is still to use slow and naive CPU algorithms [3, 4]. (We would also like to refer to the follow-up comment of reviewer dDtS.)
> > >
> > > 3) In principle, it would indeed be possible to implement the circuit and neural network on the same FPGA. However, even executing only the circuit part remains challenging on FPGAs. Existing works have only succeeded at accelerating small circuits on FGPAs (circuit of under 100 thousand nodes [1, 2], while KLay can handle over 100M nodes). So to the best of our knowledge, running both a large-scale circuit and large neural network on a single FPGA is still out of reach, and a split between neural nets on the GPU and circuits on the FPGA would be required.
> > >
> > > 4) There are further practical problems with using FPGAs for neurosymbolic learning. For example, the existing FPGA implementations only handle inference and do not consider backpropagation (KLay does not face this issue, as it can simply use the auto-differentiation of deep learning libraries). Another issue is that FPGAs remain considerably more expensive compared to consumer GPUs. Although we hope that these problems may be alleviated in the future, they currently still form significant barriers.
> > >
> > > [1] Weber, Lukas, et al. "Exploiting High-Bandwidth Memory for FPGA-Acceleration of Inference on Sum-Product Networks." 2022 IEEE International Parallel and Distributed Processing Symposium Workshops (IPDPSW). IEEE, 2022.
> > >
> > > [2] Choi, Young-kyu, et al. "FPGA acceleration of probabilistic sentential decision diagrams with high-level synthesis." ACM Transactions on Reconfigurable Technology and Systems 16.2 (2023): 1-22.
> > >
> > > [3] Manhaeve, Robin, et al. "Deepproblog: Neural probabilistic logic programming." Advances in neural information processing systems 31 (2018).
> > >
> > > [4] Huang, Jiani, et al. "Scallop: From probabilistic deductive databases to scalable differentiable reasoning." Advances in Neural Information Processing Systems 34 (2021): 25134-25145.

---

> > > > ### Author Response · Authors · 2024-11-19
> > > >
> > > > > As far as I understand, it is implied that parallel DAG traversal on GPU is a much greater challenge than parallel tree traversal. It would be helpful if you can confirm if this is true and elaborate.
> > > >
> > > > Indeed, parallelizing DAG traversal is comparatively more difficult than tree traversal. In a tree, the children of a node can be evaluated completely independently by different threads, while in a DAG inter-thread communication is necessary. Furthermore, as nodes in a DAG may be reused many times, bank conflicts arise between the warps on a GPU. For these reasons, some works have attempted to efficiently evaluate DAGs by first trying to map them to a tree (see e.g. [1]).
> > > >
> > > > [1] Shah, Nimish, Wannes Meert, and Marian Verhelst. "DPU-v2: Energy-efficient execution of irregular directed acyclic graphs." 2022 55th IEEE/ACM International Symposium on Microarchitecture (MICRO). IEEE, 2022.
> > > >
> > > > > The authors made a good point of controllability and interpretability. Out of curiosity, do the authors think if applying this technique on, for example, Large Language Models, can help solving the hallucination problem?
> > > >
> > > > The hallucination problem of language models is still an active research area and not fully solved, but neurosymbolic techniques can help mitigate it to some degree. The most basic solution is to use the language model as an information retrieval model such that it provides a verifiable source (also known as retrieval augmented generation). In relation to language models, neurosymbolic techniques have also been used successfully in e.g. toxicity prevention [1].
> > > >
> > > > [1] Ahmed, K., Chang, K. W., & Van den Broeck, G. (2024). A pseudo-semantic loss for autoregressive models with logical constraints. Advances in Neural Information Processing Systems, 36.

---

> > > > > ### Comment · Reviewer_EziM · 2024-11-19
> > > > > **Follow-up discussions (cont.)**
> > > > >
> > > > > Thank you for your revision in terms of accelerators and your detailed response.
> > > > > Please allow me to state my final concerns:
> > > > >
> > > > > 1. Is it correct that the main contribution of this work is to accelerate DAG traversal on GPU with various node types?
> > > > > 2. If that is the case, are there other DAG traversal approaches on GPUs and are they compatible to the alternating node types?

---

> > > > > > ### Author Response · Authors · 2024-11-20
> > > > > >
> > > > > > > Is it correct that the main contribution of this work is to accelerate DAG traversal on GPU with various node types?
> > > > > >
> > > > > > We stress that our definition of a circuit is precisely a DAG combined with 2 node operations (lines 57-62), as is standard in the literature. So our work indeed focuses on accelerating circuits (i.e. DAGs) on the GPU. Our main contributions towards this are 1) KLay, a layerized data structure for circuits, 2) an algorithm to efficiently evaluate KLay on the GPU, and 3) an algorithm to transform arbitrary circuits into KLay. We also clearly summarize these contributions in the first paragraph of the conclusion.
> > > > > >
> > > > > > > If that is the case, are there other DAG traversal approaches on GPUs and are they compatible to the alternating node types?
> > > > > >
> > > > > > As discussed in the related work (see lines 312), previous works have had seen limited success with accelerating circuits (i.e. DAGs) and did not demonstrate significant performance improvements of GPU execution compared to CPU evaluation (see e.g. [1,2]). On the other hand, KLay can outperform CPU baselines by over 10x in terms of runtime. We also compare with the work of [3], which we outperform by multiple orders of magnitude in our experiments. If there is any relevant existing work on parallel DAG execution we missed, we would be happy to hear about it.
> > > > > >
> > > > > > [1] Nimish Shah, Laura I Galindez Olascoaga, Wannes Meert, and Marian Verhelst. Acceleration of probabilistic reasoning through custom processor architecture. In 2020 Design, Automation & Test in Europe Conference & Exhibition (DATE), pp. 322–325. IEEE, 2020.
> > > > > >
> > > > > > [2] Md Vasimuddin, Sriram P Chockalingam, and Srinivas Aluru. A parallel algorithm for Bayesian network inference using arithmetic circuits. In 2018 IEEE International Parallel and Distributed Processing Symposium (IPDPS), pp. 34–43. IEEE, 2018.
> > > > > >
> > > > > > [3] Meihua Dang, Pasha Khosravi, Yitao Liang, Antonio Vergari, and Guy Van den Broeck. Juice: A julia package for logic and probabilistic circuits. In Proceedings of the AAAI Conference on Artificial Intelligence, pp. 16020–16023, 2021.

---

> > > > > > > ### Comment · Reviewer_EziM · 2024-11-20
> > > > > > >
> > > > > > > Thank you for the clarification. The scores are modified.

---

### Official Review · Reviewer_HTjk · 2024-11-03

**Soundness:** 3
**Presentation:** 3
**Contribution:** 3
**Rating:** 6
**Confidence:** 3

**Summary:**

In this paper, the authors introduce a new data structure called knowledge layers (KLAY) designed to enhance the parallelization of Boolean circuit computation in neurosymbolic AI. They propose organizing d-DNNF circuits into layers and applying Merkle hashing for node deduplication, subsequently reducing circuit evaluation to indexing and scattering operations. Empirical results demonstrate that KLAY achieves speedups of several orders of magnitude compared to related works, providing a significantly more efficient implementation on GPUs.

**Strengths:**

1. The utilization of modern AI accelerators is crucial for advancing the field of neurosymbolic AI, enabling the development of large-scale models and tackling more complex problems that have been previously out of reach.
2. The ability to enable parallel computation on GPUs results in speedups of multiple orders of magnitude over current state-of-the-art approaches.

**Weaknesses:**

1. The background introduction to neurosymbolic AI is somewhat lacking, providing insufficient context.
2. The experimental comparisons with state-of-the-art methods appear limited in scope compared to those discussed in related works.

**Questions:**

1. Although the title of Section 2 is "A Brief Primer on Neurosymbolic AI," it does not adequately explain the concept. It focuses primarily on the use of Boolean circuits in neurosymbolic AI and the challenges associated with their execution on GPUs. The significance and value of the work need to be more emphasized.
2. In Figure 6, there is a noticeable increase in slope when the number of instances exceeds 100. Does this behavior align with the authors' expectations regarding the linear time complexity of the algorithm?
3. In Table 1, a batch size of 1 is used. Is this a standard measurement practice? Could this choice potentially disadvantage non-parallel methods due to lower resource utilization?

---

> ### Author Response · Authors · 2024-11-17
> **Rebuttal**
>
> > The experimental comparisons with state-of-the-art methods appear limited in scope compared to those discussed in related works.
>
> We interpreted this comment in two possible way:
> 1) We did not compare to enough competing methods
> 2) We did not perform the comparison on enough problems.
>
> For the first interpretation, we believe to have included all state-of-the art methods. If the reviewer has a method in mind that we might have overlooked we would kindly ask for a specific pointer.
>
> As for the second interpretation of the comment, we believe to have an experimental evaluation with appropriate scope to show that our premise holds in general. That is, using KLay yields enormous speed-ups in run time.
>
>
> > Although the title of Section 2 is "A Brief Primer on Neurosymbolic AI," it does not adequately explain the concept. It focuses primarily on the use of Boolean circuits in neurosymbolic AI and the challenges associated with their execution on GPUs. The significance and value of the work need to be more emphasized.
>
> The contribution of our work lies in circuit evaluation, and hence this is what the background material focuses on. We have revised the title of Section 2 to “Arithmetic Circuits and Neurosymbolic AI” to more accurately reflect its contents. We have also added an example in Appendix of how neural networks can be used in combination with circuits, to motivate our paper to a wider audience.
>
> > In Figure 6, there is a noticeable increase in slope when the number of instances exceeds 100. Does this behavior align with the authors' expectations regarding the linear time complexity of the algorithm?
>
> Yes, this is expected. For small circuits, the GPU is heavily underutilized and increasing the circuit can just use additional parallelization, but at some point the GPU will be used near capacity and this slope will increase. We note that all considered algorithms have the same linear complexity, and hence for sufficiently large experiments the slope is expected to be the same.
>
> > In Table 1, a batch size of 1 is used. Is this a standard measurement practice? Could this choice potentially disadvantage non-parallel methods due to lower resource utilization?
>
> A batch size of 1 is the most challenging to parallelize as there is no data parallelism to exploit. It also occurs in practice as many neural network evaluations may be needed for the input of a single circuit evaluation. For example, batch size of 1 for the circuit of the Warcraft dataset still requires a batch of 144 for the neural network. Finally, note that we repeat Table 1 with a batch size 128 in Table 3.

---

### Official Review · Reviewer_dDtS · 2024-11-04

**Soundness:** 3
**Presentation:** 3
**Contribution:** 3
**Rating:** 8
**Confidence:** 4

**Summary:**

This paper proposes a new data structure, called knowledge layers (KLay), that allows the efficient parallelization of arithmetic circuits, which appear to be a major computational bottleneck in a specific type of neuro-symbolic AI approach.

**Strengths:**

First, the paper tackles an important challenge in neuro-symbolic AI, where scalability is still one of the major roadblocks for their general application. The achieved speed-ups of the proposed KLay on both CPU and GPU seem very promising.  Finally, the paper is well-written, and the methods are explained intuitively thanks to the use of appropriate figures and examples. This makes the paper accessible despite the methodology being quite involved.

**Weaknesses:**

1) Although the paper indicated in 3rd paragraph that it focused on a "particular flavor of neurosymbolic AI", i.e., a neural network feeding into probabilistic inference based on arithmetic circuits, it would be beneficial to reflect it explicitly in other parts, e.g., in the abstract, or a more specific title (Accelerating Arithmetic Circuits in Neurosymbolic AI), to be able to attract the right audience.

2) The actual "interface" and details between the neural network and the used arithmetic circuits remain largely a secret for readers(of course there are pointers to prior arts). It would be beneficial to open up and explain how exactly a neural network is interfaced to the arithmetic circuits, what are the assumptions and domain knowledge etc. at least for one of the tasks (e.g. MNIST addition) in an appendix.

**Questions:**

Answering the following questions/remarks could further improve the paper:

1) Unclear impact on end-to-end neuro-symbolic approaches. While the introduction clearly states the dominance of the “symbolic” part, this dominance is not obvious in the experimental results. To my understanding, only the MNIST addition dataset tests the end-to-end timing results (including "neural"+"symbolic"). What are the contributions of the individual blocks? It would be great to quantify. Moreover, this is an example where the “neural” part is rather lightweight (simple digit recognition). How would this relation between “neural” and “symbolic” would look like in other tasks?

2) Explosion in the number of KLay Nodes on “Sudoku” and “HMLC”. On these two datasets, the number of KLay nodes increases by 1.5-2.7x. What is the reason for the increase? Could you please elaborate on it?

3) Insufficient introduction of datasets. It would be good to introduce all the datasets in the appendix so that the paper becomes more self-contained. In particular, the number of instances in Figure 6 is not introduced. Are they variables, clauses, or their combination? The mentioned range over 5 orders of magnitude is not visible in Figure 6.

---

> ### Author Response · Authors · 2024-11-17
> **Rebuttal**
>
> > Although the paper indicated in 3rd paragraph that it focused on a "particular flavor of neurosymbolic AI", i.e., a neural network feeding into probabilistic inference based on arithmetic circuits, it would be beneficial to reflect it explicitly in other parts, e.g., in the abstract, or a more specific title
>
> The first sentence of the abstract explains what flavor of neurosymbolic AI we use: “A popular approach to neurosymbolic AI involves mapping logic formulas to arithmetic circuits (computation graphs consisting of sums and products) and passing the outputs of a neural network through these circuits”.
>
> > The actual "interface" and details between the neural network and the used arithmetic circuits remain largely a secret for readers(of course there are pointers to prior arts). It would be beneficial to open up and explain how exactly a neural network is interfaced to the arithmetic circuits, what are the assumptions and domain knowledge etc. at least for one of the tasks (e.g. MNIST addition) in an appendix.
>
> The inputs to the circuit are the output probabilities of neural networks (lines 107-125), forming the interface. This is an established approach in neurosymbolic AI [1, 2, 3, 4]. The contribution of our work lies in circuit evaluation, and hence the background and discussion focuses on circuits. We added a further section in Appendix that exemplifies how the MNIST-addition experiment combines circuits with neural networks. We hope this makes the paper accessible to a wider audience.
>
> [1]: Xu, Jingyi, et al. "A semantic loss function for deep learning with symbolic knowledge." International conference on machine learning (2018).
>
> [2]: Manhaeve, Robin, et al. "Deepproblog: Neural probabilistic logic programming." Advances in neural information processing systems 31 (2018).
>
> [3]: Maene, Jaron,et al. "On the Hardness of Probabilistic Neurosymbolic Learning." Forty-first International Conference on Machine Learning (2024).
>
> [4]: Ahmed, Kareem, et al. "Semantic probabilistic layers for neuro-symbolic learning." Advances in Neural Information Processing Systems 35 (2022).
>
> > Unclear impact on end-to-end neuro-symbolic approaches. While the introduction clearly states the dominance of the “symbolic” part, this dominance is not obvious in the experimental results. To my understanding, only the MNIST addition dataset tests the end-to-end timing results (including "neural"+"symbolic"). What are the contributions of the individual blocks? It would be great to quantify. Moreover, this is an example where the “neural” part is rather lightweight (simple digit recognition). How would this relation between “neural” and “symbolic” would look like in other tasks?
>
> The circuit evaluation time typically dominates the inference in neurosymbolic experiments. For instance, when running the MNIST-addition experiment with SoTA methods (e.g. Scallop or DeepProbLog), the neural network takes less than 1% of the inference runtime. For reference, we have added the neural network inference timings for the considered neurosymbolic tasks in Table 1 and 3. Although the precise ratio between circuits and neural nets depends on the specific experiment setup, the absolute improvement in execution time of KLay is very significant as long as the circuit is large (see Figure 6), regardless of the choice of neural net.
>
> > Explosion in the number of KLay Nodes on “Sudoku” and “HMLC”. On these two datasets, the number of KLay nodes increases by 1.5-2.7x. What is the reason for the increase?
>
> The layerization procedure of KLay introduces new nodes such that each node only has children in the previous layer (see Section 3.1 or a visualization in Figure 2). Depending on the circuit structure, this can be a considerable amount of new nodes. However, our experiments show that KLay is much faster despite this increase in the number of nodes (see Appendix B).
>
> > Insufficient introduction of datasets. It would be good to introduce all the datasets in the appendix so that the paper becomes more self-contained. In particular, the number of instances in Figure 6 is not introduced. Are they variables, clauses, or their combination?
>
> For the synthetic experiment in Figure 6, there are 110 instances (SDD circuits). These are generated by compiling random 3-CNF formulas (see line 400). We have added additional details on how the synthetic experiments are generated in Appendix D.
>
> > The mentioned range over 5 orders of magnitude is not visible in Figure 6.
>
> The sizes of the circuits in Figure 6 vary between 1000 nodes to 100 million nodes. This is visualized in Figure 7 (left).

---

### Meta-Review · Area_Chair_jBBj · 2024-12-21

**Metareview:**

This paper a new data structure called knowledge layers (KLAY) to represent arithmetic circuits that can be efficiently parallelized on GPUs, in order to address the issue that arithmetic circuits are challenging to run on modern tensor accelerators. All the reviewers, which I also agree from my own reading, that the idea is novel and interesting. I would recommend acceptance due to follow reasons:

1. Reasons to accept:

(1) Novelty and important contribution: It tackles an very important challenge in neuro-symbolic AI. The achieved speed-ups of the proposed KLay on both CPU and GPU seem very promising. The utilization of modern AI accelerators is crucial for advancing the field of neurosymbolic AI. Empirical results are displayed in a clear way, showing the effective superiority of the proposed method over existing ones.

(2)  Well-written Paper: The methods are explained clearly with appropriate figures and examples, and the whole paper is easy to follow.

2. Weaknesses of the paper

The background, in particular symbolic AI and motivation is not sufficient. But after rebuttal, this has been improved.

Overall, I believe this is a solid paper with novel contributions, and would like to recommend acceptance. Meanwhile, I sincerely hope that the authors could revise their paper based on discussions in the rebuttal.

**Additional Comments On Reviewer Discussion:**

Originally reviewer HTjk and EziM concerned the writing but in the rebuttal, the authors have revised and addressed these concerns.

---

### Decision · Program_Chairs · 2025-01-22

Accept (Poster)